# Histopathological Grading, Clinical Staging and CD 31 Expression of Canine Splenic Hemangiosarcoma

**DOI:** 10.3390/vetsci10030190

**Published:** 2023-03-02

**Authors:** Ka-To Chu, Omid Nekouei, Jeanine Rhoda Sandy

**Affiliations:** Jockey Club College of Veterinary Medicine and Life Sciences, City University of Hong Kong, Hong Kong 999077, China

**Keywords:** canine splenic hemangiosarcoma, prognosis, grading, CD 31

## Abstract

**Simple Summary:**

Canine hemangiosarcoma is an aggressive cancer with a short median survival time, commonly found in the spleen of older dogs. There are limited tests available to provide clinically useful prognostic information to guide therapies once a diagnosis is made. Predictors of survivability investigated in this case series included the application of a previously published tumor histological grading scheme, histological assessment of tumor cell atypia, clinical staging, and the level of CD 31 expression by tumor cells in 16 dogs diagnosed with splenic hemangiosarcoma. Medical records were reviewed, the time of death was obtained, and survival data were analyzed statistically. Histopathological grading, cellular atypia, and clinical staging of canine splenic hemangiosarcomas were not significantly associated with the median survival time of the study dogs. Strong expression of CD 31 by canine splenic hemangiosarcoma tumor cells was observed in dogs with short survival times. This study provides a basis for future work to further investigate the prognostic usefulness of CD 31 expression for determining the survivability of canine splenic hemangiosarcoma patients.

**Abstract:**

Canine hemangiosarcoma is a common, highly fatal tumor of older dogs, and predictors of survivability may provide clinically useful information. The objectives of this case series were to determine if a previously published tumor histological grading scheme, the level of tumor cellular atypia, clinical staging, or the level of CD 31 expression were useful for predicting the survival time in dogs with splenic hemangiosarcoma. Canine splenic hemangiosarcomas from 16 dogs were histologically graded, clinically staged, and assessed for CD 31 expression. Medical records were reviewed, the date of death was obtained, and survival data were analyzed statistically. Histopathological grading and clinical staging of canine splenic hemangiosarcomas, and the expression of CD 31 by the tumor cells were not significantly associated with the median survival time of the dogs in this study. However, strong expression of CD 31 by canine splenic hemangiosarcoma tumor cells was observed in dogs with short survival times, which warrants further studies to evaluate the potential prognostic value of CD 31 expression for the survival of dogs with splenic hemangiosarcoma.

## 1. Introduction

Canine hemangiosarcoma (HSA) is an aggressive cancer commonly found in older dogs, and splenic HSA is the commonest form of the disease, representing approximately half of the HSA cases in dogs [1,2,3,4]. Median survival times (MST) of dogs treated with splenectomy alone is less than 3 months [3,5,6]. To date, the most prognostically useful factor for canine hemangiosarcoma is the clinical staging of the disease, with dogs at stage I having longer survival times compared with dogs at stages II and III [3,6,7,8,9], but the usefulness of clinical staging is somewhat disputed in the literature, [10,11,12] with the administration of chemotherapy post splenectomy generally accepted as the most useful factor to extend survival time [5,6,7,9].

Histological grading of canine HSA was first proposed by Ogilvie et al. 1996 [7] but is not routinely performed as there little data to support its prognostic value. However, tumors exhibiting increased mitotic counts may be associated with shortened survival times [13]. Immunohistochemistry (IHC) is not routinely required by veterinary pathologists to reach a diagnosis of canine splenic HSA but may be useful to identify poorly differentiated tumors or epithelioid variants of HSA using endothelial markers such as CD 31, Factor VIII, and von Willebrand Factor (vWF) [14,15,16,17]. CD 31, or platelet endothelial cell adhesion molecule-1 (PECAM), is an adhesion receptor expressed by endothelial cells involved in regulating apoptosis and maintaining vascular permeability and integrity [18,19]. Canine HSA tumor expression of these endothelial markers has not been specifically investigated for their usefulness in predicting post-splenectomy survival although dogs with less differentiated tumors may fail to express these markers [17,20,21]. Cellular atypia was assessed as per the scoring scheme of Maharani et al., 2018 [21].

In this study, clinical stage, histological grade, cellular atypia, and the expression level of CD 31 were determined in 16 dogs with splenic HSA, and their potential associations with the survival time of dogs post splenectomy were evaluated. In addition, the survival times of dogs were compared based on each individual histological feature of the published grading scheme of Ogilvie et al. 1996 [7] (Table 1), which included mitotic count and tumor differentiation.

## 2. Materials and Methods

### 2.1. Animals and Study Design

All canine splenic HSA cases from 2018 to 2021 were retrieved from the electronic medical record database of City University of Hong Kong’s Veterinary Diagnostic Laboratory (VDL). Dogs were included in this study based on the following selection criteria: splenectomized dogs with a histological confirmation of splenic HSA, successfully discharged from hospital following splenectomy as their sole source of treatment, with the time of death recorded. Dogs excluded from this study were those who received adjunct chemotherapy post splenectomy, which can introduce a confounding effect on survivability, as various treatment regimens are known to improve survival times [3,5,6,7,9], and we intended to examine the potential effect, if any, of CD 31 expression by tumor cells on post splenectomy survival times.

Splenectomized HSA dogs that died or were euthanized before hospital discharge and dogs whose clinical outcomes were lost to follow-up were also excluded from this study. Sixteen dogs met the eligibility criteria, and the patient’s sex, breed, and age at the time of splenic HSA diagnosis were recorded (Table 2). The survival time for each dog was determined from the clinical records and defined from the date of splenectomy to the date of death. Clinical staging at the time of splenectomy was performed for all dogs based on abdominal ultrasound, echocardiogram, and/or 3-view thoracic radiographs, as well as visual assessment of abdominal organs during surgery. The previously established three-tiered staging system (stages I–III) for canine patients with HSA was used, based on the size of the primary tumor, evidence of tumor rupture, and presence of regional lymph node and/or distant metastasis [22].

### 2.2. Histology

Hematoxylin and eosin (H&E)-stained glass slides of splenic tissue, previously fixed in 10% neutral buffered formalin and embedded in paraffin cassettes, were retrieved for each dog. For each case, 5–7 slides were originally created to confirm a diagnosis of splenic HSA as recommended by Herman et al. 2019 [23]. For each dog, a single slide was identified as being most representative of the tumor cell population and this slide was chosen for histological grading and assessment of cellular atypia, with the matching paraffin block retrieved from storage for IHC staining.

Each H&E glass slide was graded as per Ogilvie et al. 1996 [7], summarized in Table 1. In brief, four histological parameters for neoplastic endothelial cells were assessed: cellular differentiation, nuclear pleomorphism, mitotic count, and percentage of necrosis. Cellular differentiation is related to the ability of neoplastic endothelial cells to form recognizable vessels. Nuclear pleomorphism equated to uniformity of nuclear size and shape between tumor cells and represented a measure of cellular atypia. Mitotic count was calculated by counting all mitotic figures in 10 high-power fields exhibiting the highest mitotic activity. As the microscope had an ocular field number of 22, the total area evaluated was 2.37 mm^2^; mitotic count/2.37 mm^2^. Tumor necrosis was histologically defined by abundant cell debris, loss of tissue architecture, and/or loss of H&E tinctorial staining intensity. Each factor combined to create a final score equating to the three-tiered histological grade. Grading was performed independently by the first (K.T.C.) and last author (J.R.S.). If grades differed, repeat grading was performed by both authors simultaneously, so that a consensus score was reached, which was then recorded as a data point. Cellular atypia was scored as per Maharani et al., 2018 [21]. Neoplastic endothelial cells were given a score of 1 if the tumor cells were spindle-shaped with an elongated nucleus (width less than 1 erythrocyte). Score 2 cells were spindle-shaped with a plump oval nucleus (width of approximately 1–2 erythrocytes). Score 3 cells were polygonal- or cuboidal-shaped with plump cytoplasm, and a large round or irregularly shaped nucleus in which three or more erythrocytes could easily fit.

### 2.3. Immunohistochemistry

CD 31 IHC was performed on the same tissue section used for grading, using the Bond-III Fully Automated IHC and ISH Staining System (Leica Biosystems, Newcastle, United Kingdom). A 4 µm section was cut from each paraffin block, deparaffinized with descending concentrations of alcohol and xylene, and rinsed in tap water followed by distilled water. Heat-induced epitope retrieval was performed using the BOND Epitope Retrieval Solution 2 (ER2) (EDTA, pH 9.0; Leica Biosystems, Newcastle, UK) at 100 °C for 30 min. Tissue samples were washed 3 times with Bond Wash Solution (Leica Biosystems, Newcastle, UK). For blocking, sections were incubated in hydrogen peroxide for 10 min at room temperature (RT), followed by 3 repeat washings with Bond wash solution. Sections were then incubated at RT with anti-CD 31 in a 1:30 working solution (Monoclonal Mouse Anti-Human CD 31, Endothelial Cell, Cide M0823, Agilent Technologies, Santa Clara, CA, USA) for 15 min, and sections were washed 3 times with the Bond Wash Solution. The 3,3’Diaminobenzidine (DAB) detection system was used as the chromagen. Sections were treated with a post primary solution for 8 min at RT, washed with Bond wash 3 times for 2 min each then treated with the polymer solution for 8 min, washed for 2 min in Bond wash 3 times, followed by 3 2-min washes with distilled water. The final steps were treatment with DAB solution for 10 min, followed by 3 washes with distilled water, then counterstaining with hematoxylin. For each case, a positive and negative control tissue sample was included. Positive controls represented paraffin-embedded tissues from previously diagnosed cases of canine vascular neoplasms, including hemangioma or well-differentiated HSA, prepared under the same staining protocol to ensure identical staining conditions. A negative control slide was produced with the same protocol on HSA tissue, with the exclusion of the primary anti-CD 31 antibody.

Tissue from one dog (dog 2), with a poorly differentiated splenic HSA, had vWF and microglia (Iba1) IHC performed on representative splenic tissue during the initial diagnostic workup of the case. For both of these antibodies, the same protocol was utilized as for CD 31 with the following exceptions. Iba1 (BIOCARE Medical, Pacheco, CA, USA) heat epitope antigen retrieval was performed for 30 min using Epitope Retrieval Solution 1 (ER1) (citrate, pH 6.0; Leica Biosystems, Newcastle, UK). Blocking with hydrogen peroxide was performed for 5 min at RT and the primary antibody was used as a 1:33 working solution incubated for 30 min at RT and vWF (DAKO Omnis, Agilent, Santa Clara, CA, USA) as a 1:200 working solution. Heat epitope antigen retrieval was performed for 20 min using ER2 (EDTA, pH 9.0; Leica Biosystems, Newcastle, UK). Blocking with hydrogen peroxide was performed for 10 min at RT.

Positive expression of CD 31 by tumor cells was characterized by either brown DAB staining of the cell membrane or both the cell membrane and cytoplasm. Expression was scored from 0 to 3 (0 = no brown staining, 1 = pale brown membranous staining with minimal/no cytoplasmic staining, 2 = coarse mid/dark brown stippling of the cytoplasm with or without membranous staining, 3 = diffuse dark brown cytoplasmic/membranous staining of tumor cells), based on the protocol of Maharani et al., 2018. [21]. Positive control tissues were included and endothelial cells near the tumor mass were utilized as internal positive controls.

### 2.4. Statistical Analysis

Due to the low number of dogs in this study and their imbalanced distribution within some categories of the predictor variables of interest, all predictors were dichotomized to enable more robust screening for potential statistical associations between each variable and the survival time as the outcome of interest (Table 3). Within each category of predictors, median survival time (MST) and the corresponding range were calculated and compared using the non-parametric “equality-of-median test” in Stata v17 (StataCorp, College Station, TX, USA). A significance level of 0.05 was considered in all statistical tests. Two Fisher’s exact tests were conducted to determine whether clinical stage and histological grade had any associations with the intensity of CD 31 expression by tumor cells.

## 3. Results

### 3.1. Demographics

Demographic information, clinical staging, histological grading, intensity of CD 31 expression, and survival time of canine splenic HSA patients post splenectomy are summarized in Table 2. Three Labradors, three schnauzers, two corgis, two bichon frises, two poodles, and one each of beagle, golden retriever, and dachshund and one mongrel dog were included (Table 2). Twelve dogs were male and four were female. The age of dogs ranged from 8 to 14 years, with a median of 11 years. For all dogs, the survival time ranged from 5 to 201 days, with a median of 68 days (Figure 1; Table 2).

### 3.2. Clinical Staging

Clinical staging classified patients as follows: two dogs (dogs 2 and 8) were stage I with survival dates of 10 and 67 days, respectively, seven dogs (dogs 5, 9, 11, 12, 13, 15, 16) were stage II with survival dates between 19 and 201 days, and seven dogs (dogs 1, 3, 4, 6, 7, 10, 14) were stage III with survival dates between 5 and 129 days (Table 2). Eight dogs (dogs 2, 5, 6, 7, 8, 10, 12, 15) were euthanized, with the clinical reason recorded as poor quality of life with pallor and lethargy, and specifically for some dogs, severe anemia and thrombocytopenia (dog 2, stage I and dog 12, stage II), moderate anemia (dog 8, stage I), severe anemia (dog 15, stage II), a nonresectable liver mass (dog 10, stage III), and shock (dog 6, stage III). Two dogs (dogs 1 and 3, both stage III) were reported to have passed away by their owners, and details of death were not recorded for six dogs (dogs 4, 9, 11, 13, 14, 16).

### 3.3. Histological Grade and CD 31 Expression

There were seven grade 1 tumors (dogs 1, 8, 9, 10, 11, 12, 13), nine grade 2 tumors (dogs 2, 3, 4, 5, 6, 7, 14, 15, 16) and no grade 3 tumors (Table 2). Tumor cells from one dog (dog 2) failed to express CD 31, one dog (dog 6) had weak (intensity 1) CD 31 expression, six dogs (dogs 10, 11, 12, 13, 15, 16) had moderate (intensity 2), and eight dogs (dogs 1, 3, 4, 5, 7, 8, 9, 14) had strong (intensity 3) CD 31 expression (Table 2, Figure 2). Dogs with moderate CD 31 expression (intensity 2) tended to have the longest survival times irrespective of their histological grade, whilst dogs with no (intensity 0), weak (intensity 1), or strong (intensity 3) CD 31 expression tended to have the shortest survival times.

### 3.4. Cellular Atypia and CD 31 Expression

Endothelial cells from four dogs scored 1 on cellular atypia, and either intensity 2 (dogs 10, 12, 16) or 3 (dog 7) for CD 31 expression. Endothelial cells from nine dogs scored 2 on cellular atypia and either scored 1 (dog 6), 2 (dogs 11, 13, 15), or 3 (dogs 1, 3, 8, 9, 14) for intensity of CD 31 expression. Endothelial cells from three dogs scored 3 on cellular atypia and either scored 3 (dogs 4, 5) or 0 (dog 2) for intensity of CD 31 expression. 

Cellular atypia as per Maharani et al., 2018 [21]. Score 1: spindle-shaped cells with elongated nucleus (width less than 1 erythrocyte). Score 2: spindle-shaped cells with plump oval nucleus (width of approximately 1–2 erythrocytes). Score 3: polygonal- or cuboidal-shaped cells with plump cytoplasm and a large round nucleus in which three or more erythrocytes could easily fit.

Mitotic count. Total number of mitotic figures counted in 10 high power field (×40 objective) using an ocular with a field number of 22. Equates to total number of mitotic figures in 2.37 mm^2^.

CD 31 intensity. Intensity of CD 31 expression by tumor cells (brown coloration of tumor cells) on histopathology based on Maharani et al., 2018 [21]. (0 = no brown staining, 1 = pale brown membranous staining with minimal/no cytoplasmic staining, 2 = coarse mid/dark brown stippling of the cytoplasm with or without membranous staining, 3 = diffuse dark brown cytoplasmic/membranous staining of tumor cells).

Table 3 depicts the distribution of survival time (median and range) by the predictor variables of interest. None of the predictors had a statistically significant association with the survival time of all study dogs (all *p*-values > 0.1). MST for dogs with level 2 CD 31 expression intensity was 94.5 days compared with dogs with level 3 CD 31 expression, which was 28.5 days, indicating that dogs with tumor cells with level 3 (high) CD 31 expression (dogs 1, 3, 4, 5, 7, 8, 9, 14) had the lowest MST (Table 2). No statistically significant associations were observed between clinical stage, histological grade, or cellular atypia and the intensity of CD 31 expression in tumor cells of the 16 patients (*p* = 0.315 and *p* = 0.999, respectively).

## 4. Discussion

Clinical staging of 16 dogs diagnosed with splenic HSA, treated by splenectomy alone, showed no significant correlation with survival times. Histological grading of canine splenic HSA, using the scheme of Ogilvie et al. 1996 [7] and assessment of cellular atypia as per Maharani et al., 2018 [21] were not useful in predicting survival times. While dogs with tumor cells that strongly expressed CD 31 (CD 31 expression intensity of level 3) had shorter survival times compared to those with lower CD 31 intensity levels; this difference was not statistically significant. With respect to the descriptive nature of the current study (case series), sample selection process, and the limited number of cases, the results of statistical comparisons must be interpreted with caution and only considered as a “hypothesis generator” to be tested and validated in follow-up studies specifically designed to evaluate the prognostic value of the outlined predictors.

Study dogs included twelve males and four females, similar to the reported slight male predisposition for this disease [1,14,20,24]; however, the observed sex ratio (12/4) could have been a result of sample selection in this case series. The median age at diagnosis of splenic HSA in this study was 11 years, consistent with the reported age predisposition of canine splenic HSA [1,2,3,11,20]. The age of dogs in this study had a narrow distribution, so age was not expected to confound the analysis of survival times in our study.

The lack of correlation between the clinical stage of canine splenic HSA and survival time in this study is similar to previous reports [10,11,12], but is at odds with other studies [3,4,6,7,8,9,22,25], which may highlight the need for the further investigation of other, more reliable predictors of survival times. The MST of dogs in the current study was 68 days, similar to previously reported MST ranges of 19 to 86 days in dogs with splenic HSA treated by splenectomy alone [3,5,6,11]. None of the 16 dogs were alive one year after their diagnosis of splenic HSA. This finding was consistent with the short survival of dogs with visceral HSA (splenic HSA representing the majority of visceral cases), who typically die within a year despite treatment [3,7,8,13,26,27]. Half of the dogs in this study were euthanized (8/16 dogs; dogs 2, 5, 6, 7, 8, 10, 12, 15) because of poor quality of life post splenectomy, commonly associated with lethargy and anemia [4,12,22,28].

We did not observe any correlations between survival time and the individual histopathological features of neoplastic cells, including the mitotic count and degree of nuclear pleomorphism. These histopathological features have been reported to be associated with the survival time in dogs treated for canine HSA by splenectomy and chemotherapy [7,13]. However, post-splenectomy chemotherapy represents a confounding effect on survivability, as various treatment regimens are known to improve survival times [3,5,6,7,9].

Tumor cells from three dogs in this study exhibiting the most cellular atypia (score of 3) had survival times of 10, 17, and 19 days (dogs 2, 4, 5), shorter than the MST of 68 days for this study. This finding may suggest that increasing levels of cellular atypia could represent a negative prognostic indicator for dogs with splenic HSA. When cellular atypia was linked with CD 31 expression, no clear pattern was seen to support the finding of Maharani et al., 2018 [21], where loss of the intensity of CD 31 expression by tumor cells was seen with increasing cellular atypia. Eight dogs (dogs 1, 3, 4, 5, 7, 8, 9, 14) with level 3 CD 31 expression intensity had a median survival time of 28.5 days, compared with a median survival time of median 94.5 days for six dogs (dogs 10, 11, 12, 13, 15, 16) with level 2 CD 31 expression intensity. Normal canine endothelial cells in control tissues for this study expressed CD 31 at levels 2 and 3, similar to previous studies [20], so a negative correlation of level 3 (strong) expression of CD 31 by neoplastic splenic endothelial cells with survival time in dogs post splenectomy is an interesting and unexpected finding. Increased expression of CD 31 may impart properties to neoplastic endothelial cells, aiding malignant behavior due to its role as a regulator of endothelial apoptosis. Upregulation of various cellular receptors such as CD 44, HER-2, HER-2 in malignant canine mammary tumors [29] and gain of C-MYC function in canine prostatic carcinomas [30] have been linked to increased invasiveness and metastatic potential of these tumors. Perhaps maintenance of cellular receptors such as CD 31 at levels similar to nonneoplastic endothelial cells imparts a survival tactic by neoplastic cells. Therefore, the influence of both cellular atypia and CD 31 expression as potential prognostic indicators for dogs with splenic HSA warrants further testing in future, well-designed studies.

The loss or reduction in CD 31 expression by two dogs in our study (dogs 2 and 6) is also of interest, as each dog had very short survival times of 10 and 19 days, respectively, post splenectomy. Absent or weak expression of various differentiation markers by tumor cells is reported in different types of canine tumors where it can represent a negative prognostic finding with an increased risk of invasive and metastatic disease [30,31,32]. Maharani et al., 2018 [21] showed that pleomorphic visceral HSA cells from 17 dogs had decreased expression of CD 31 and vWF, and others have reported rare canine HSAs with loss of CD 31 expression [16,17,20] but expression of CD 31 has not been previously correlated with survival times. Venkataramani et al. 2018 [33] assessed CD 31 expression within tissue cultures derived from angiosarcomas, the human counterpart to animal hemangiosarcomas. They found that tumors expressing CD 31^LOW^ patterns represented prognostically worse tumors when compared with tumors expressing CD 31^HIGH^. Therefore, maintenance or over-expression, as well as reduction or loss of cellular receptor expression such as CD 31, may both impart different growth advantages to neoplastic endothelial cells, but further studies are needed to determine their prognostic usefulness. In addition, correlating the success rate that chemotherapeutic agents extend the survival times of dogs with splenic HSA with tumor expression of CD 31 would be of interest, as CD 31^LOW^ endothelial cells in human angiosarcomas has been linked to resistance to various chemotherapeutic agents, including doxorubicin, which is used in the treatment of canine HSA [33].

Although there were over 200 cases of splenic HSA cases in our database, the majority of dogs were excluded from our study due to the loss of clinical follow-up post splenectomy. Another limitation of our study, common to most studies investigating survival times in dogs with HSA, was the lack of postmortem data, so that the death of these dogs was assumed to be attributed solely to their clinical diagnosis of splenic HSA. A final limitation in this study involves problems with repeatability when applying the grading schemes discussed in this paper. There is a high likelihood of generating data with considerable inter-pathologist variation, especially when scoring the intensity of CD 31 expression by tumor cells, which may limit the ability to differentiate between CD 31 intensity levels 2 and 3.

## 5. Conclusions

Clinical staging, histological grading, cellular atypia, and individual histological criteria provided no prognostic information in terms of survivability post splenectomy for the 16 canine HSA patients studied in this case series, which may not be reflective of the general population due to the small number of cases and the case selection process. Over- or underexpression of CD 31 may provide useful information in predicting the survival time in dogs with HSA post splenectomy. Further studies with larger sample sizes are warranted to specifically evaluate the expressivity of CD 31 by the tumor cells, and its prognostic value, either from splenectomized patients or possibly using splenic aspirates. CD 31 expression could also be assessed to examine the role that the loss of this marker may have in chemotherapeutical treatment failures of canine splenic HSA.

## Figures and Tables

**Figure 1 vetsci-10-00190-f001:**
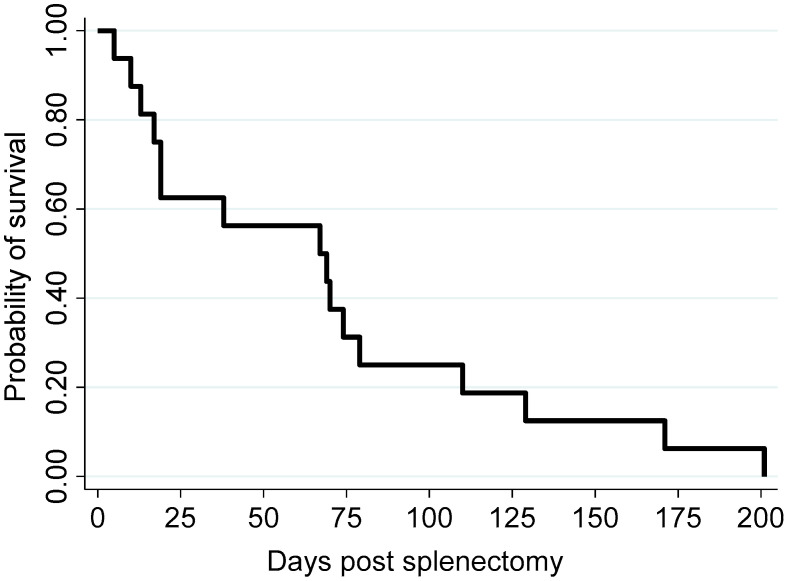
Kaplan–Meier survival curve illustrating the survival probability of the 16 study dogs following splenectomy.

**Figure 2 vetsci-10-00190-f002:**
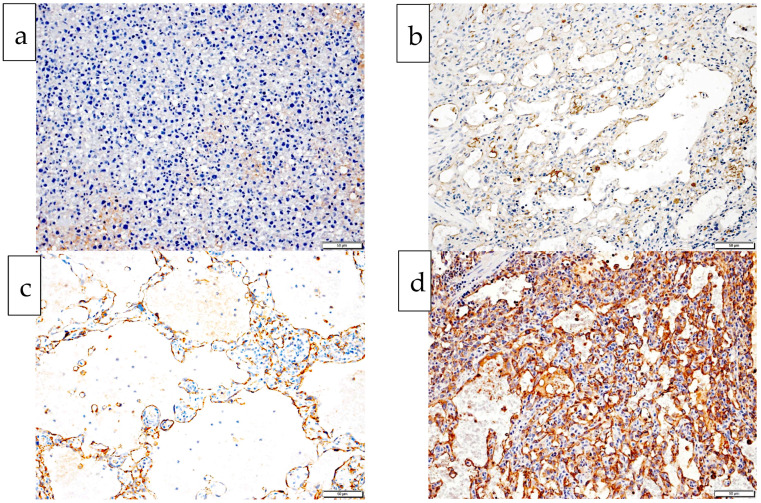
Histological images from four dogs with splenic hemangiosarcoma exhibiting each of the four intensity levels of endothelial CD 31 expression (0 to 3). Chromogen is DAB, counterstained with hematoxylin. (**a**) Endothelial cells fail to express any CD 31. There is a small amount of patchy pale brown background staining: CD 31 intensity 0, dog 2. (**b**) Endothelial cell membrane is stained pale brown with minimal/no cytoplasmic staining: CD 31 intensity 1, dog 6. (**c**) Endothelial cell cytoplasm show coarse mid/dark brown stippling of the cytoplasm with or without membranous staining: CD 31 intensity 2, dog 13. (**d**) Endothelial cells show diffuse dark brown cytoplasmic/membranous staining: CD 31 intensity 3, dog 14.

**Table 1 vetsci-10-00190-t001:** Canine splenic hemangiosarcoma grading scheme from Ogilvie et al. 1996 [7].

Differentiation	Score
Well differentiated: numerous irregular vascular channels predominate in all fields	1
Moderately differentiated: ≥50% of the tumor has well-defined vascular channels	2
Poorly differentiated: solid sheets of spindle cells with few vascular channels	3
Nuclear pleomorphism	Score
No difference in nuclear size and shape	0
Minimal variation	1
Moderate variation (2× size difference)	2
Marked variation (>2× size difference)	3
Mitotic count (in 10 high power [×40 objective] fields in 2.37 mm^2^)	Score
0 to 10	0
11 to 20	1
21 to 30	2
>30	3
Tumor necrosis	Score
No necrosis	0
<25%	1
25% to 50%	2
>50%	3
Histological grade (3-graded)	Total score
Grade I	≤5
Grade II	6 to 9
Grade III	10 to 12

**Table 2 vetsci-10-00190-t002:** Demographic information at the time of splenectomy, clinical stage, cellular atypia score, histological grade, mitotic count, CD 31 intensity, and survival time post splenectomy for 16 dogs with splenic hemangiosarcoma.

Dog Number	Breed	Age (year)	Sex	Clinical Stage	Cellular Atypia	Differentiation	Nuclear Pleomorphism	Mitotic Count/2.37 mm^2^ (Mitotic Score)	Tumor necrosis	Histological Grade	CD 31 Intensity	Survival Time (days)
1	Dachshund	14	M	3	2	1	1	5 (0)	0	1	3	5
2	Corgi	8	M	1	3	3	2	21 (2)	1	2	0	10
3	Bichon Frise	9	M	3	2	2	2	17 (1)	1	2	3	13
4	Golden retriever	12	M ^a^	3	3	2	3	22 (2)	2	2	3	17
5	Mongrel	10	F	2	3	2	1	33 (3)	1	2	3	19
6	Schnauzer	14	M	3	2	2	2	11 (1)	1	2	1	19
7	Beagle	11	M	3	1	2	2	2 (0)	0	2	3	38
8	Schnauzer	13	M	1	2	1	1	2 (0)	0	1	3	67
9	Labrador	11	F	2	2	2	2	1 (0)	0	1	3	69
10	Corgi	10.5	M	3	1	2	2	1 (0)	1	1	2	70
11	Labrador	9.5	M	2	2	2	2	3 (0)	0	1	2	74
12	Toy poodle	13.5	F	2	1	1	1	15 (1)	1	1	2	79
13	Labrador	10	M	2	2	2	1	29 (2)	0	1	2	110
14	Bichon Frise	12	M ^a^	3	2	2	1	15 (1)	2	2	3	129
15	Schnauzer	11	F	2	2	2	2	26 (2)	1	2	2	171
16	Poodle	11.5	M	2	1	2	2	9 (0)	2	2	2	201

^a^ Entire animals. All other dogs were spayed/neutered. M: male. F: female.

**Table 3 vetsci-10-00190-t003:** Screening for potential associations between the median survival time (MST) and the dichotomized predictors of interest for the 16 study dogs with splenic hemangiosarcoma.

				Survival Time (days)		
	Predictor Variable	Level (Combined Categories)	Number of Dogs	Median	Range	*p*-Value ^a^
1	CD 31 intensity	0, 1, 2	8	76.5	10–201	0.132
		3	8	28.5	5–129	
3	Histopathological grade	1	7	70	5–101	0.315
		2	9	19	10–201	
4	Clinical stage	1, 2	9	74	10–201	0.314
		3	7	19	5–129	
5	Differentiation	1	3	67	5–79	0.999
		2, 3	13	69	10–201	
6	Nuclear pleomorphism	1	6	73	5–129	0.999
		2, 3	10	53.5	10–201	
7	Mitotic count	0	5	74	67–201	0.282
		1, 2, 3	11	19	5–171	
8	Tumor necrosis	0	6	68	5–110	0.999
		1, 2	10	44.5	10–201	
9	Age (year)	≤11	9	69	10–171	0.999
		>11	7	67	5–201	
10	Sex	Female	4	74	19–171	0.569
		Male	12	52.5	5–201	

^a^ *p*-values are from the non-parametric equality-of-median tests.

## Data Availability

All relevant data have been included in this submission. Additional data exist, incorporated as part of confidential medical records of the attending veterinary hospital.

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
