# Peer review of "Histopathological Grading, Clinical Staging and CD 31 Expression of Canine Splenic Hemangiosarcoma"

_vetsci, 2023, doi:10.3390/vetsci10030190_

Round 1

Reviewer 1 Report

This work is interesting, and could have had higher value with more cases.

Author Response

Dear Reviewer 1.  Many thanks for providing details of corrections for my manuscript.  All corrections, as directed by your review have been completed.

Many thanks,

Jeanine Sandy 

Reviewer 2 Report

Dear Sir/Madam,

It is my pleasure to review your case report, I included all my comments in the attached PDF file. Would you please consider all the notes I have written? This will enhance the study's originality and answer all questions. Do you think that the dogs' breed has any correlations with Histopathological grading, clinical staging, and CD31 expression?

Sincerely,

Author Response

Many thanks for the excellent feedback from reviewer 2.  I have made comments and corrections addressing each point raised by the reviewer in the editted document.  Please refer to the attached word document.

Many thanks,

Jeanine Sandy 

Reviewer 3 Report

This Case Report describes the expression of CD31 in 16 cases of canine splenic haemangiosarcoma and its correlation with patient survival, it is a modest but interesting study.

My comments are as follow:

2.3:   last sentence states that endothelial cells near the tumour mass were utilized as internal positive controls.  I find it quite surprising that there was sufficient normal tissue surrounding the tumour mass to allow this, as splenic haemagiosarcomas often represent large masses which occupy significant portions of the spleen. So question whether this was true for every case?

Figure 1, the difference between the intensity of staining in section labelled (b) in comparison with (c) is not convincingly different on the version I printed.  Please can this be addressed.

I do have some concerns about the results only becoming significant when 2 dogs were excluded from the analysis due to no / weak CD31 staining, this could be viewed as data manipulation.  However, I feel that the authors have addressed this point reasonably well in the discussion.

Author Response

Comments addressed by author for Reviewer 3.  

COMMENT 2.3:   last sentence states that endothelial cells near the tumour mass were utilized as internal positive controls.  I find it quite surprising that there was sufficient normal tissue surrounding the tumour mass to allow this, as splenic haemagiosarcomas often represent large masses which occupy significant portions of the spleen. So question whether this was true for every case?  

ANSWER: Yes, there was sufficient normal vasculature available on each histological slide stained with CD 31 from EVERY dog examined.  We chose each slide to be stained by CD 31 carefully, avoiding areas where only tumour was seen - to intentionally choose fields where normal splenic tissue was still identifiable. 

COMMENT: Figure 1, the difference between the intensity of staining in section labelled (b) in comparison with (c) is not convincingly different on the version I printed.  Please can this be addressed.

ANSWER:  image C has been replaced.  

COMMENT I do have some concerns about the results only becoming significant when 2 dogs were excluded from the analysis due to no / weak CD31 staining, this could be viewed as data manipulation.  However, I feel that the authors have addressed this point reasonably well in the discussion.

ANSWER:  The relationship of CD 31 expression and prognosis for canine splenic HSA needs further work with more dogs.  This is a case report where we asked the question, but more work needs to be done to see if expression of CD 31 and other endothelial markers may provide prognostically useful information to guide therapies and clinical decision making.  

Many thanks,

Jeanine Sandy